# Current State of Molecular and Serological Methods for Detection of Porcine Epidemic Diarrhea Virus

**DOI:** 10.3390/pathogens11101074

**Published:** 2022-09-21

**Authors:** Monika Olech

**Affiliations:** Department of Pathology, National Veterinary Research Institute, 24-100 Puławy, Poland; monika.olech@piwet.pulawy.pl

**Keywords:** porcine epidemic diarrhea virus, PEDV, molecular diagnostics, serological diagnostics

## Abstract

Porcine epidemic diarrhea virus (PEDV), a member of the *Coronaviridae* family, is the etiological agent of an acute and devastating enteric disease that causes moderate-to-high mortality in suckling piglets. The accurate and early detection of PEDV infection is essential for the prevention and control of the spread of the disease. Many molecular assays have been developed for the detection of PEDV, including reverse-transcription polymerase chain reaction (RT-PCR), real-time RT-PCR (qRT-PCR) and loop-mediated isothermal amplification assays. Additionally, several serological methods have been developed and are widely used for the detection of antibodies against PEDV. Some of them, such as the immunochromatography assay, can generate results very quickly and in field conditions. Molecular assays detect viral RNA in clinical samples rapidly, and with high sensitivity and specificity. Serological assays can determine prior immune exposure to PEDV, can be used to monitor the efficacy of vaccination strategies and may help to predict the duration of immunity in piglets. However, they are less sensitive than nucleic acid-based detection methods. Sanger and next-generation sequencing (NGS) allow the analysis of PEDV cDNA or RNA sequences, and thus, provide highly specific results. Furthermore, NGS based on nonspecific DNA cleavage in clustered regularly interspaced short palindromic repeats (CRISPR)–Cas systems promise major advances in the diagnosis of PEDV infection. The objective of this paper was to summarize the current serological and molecular PEDV assays, highlight their diagnostic performance and emphasize the advantages and drawbacks of the application of individual tests.

## 1. Introduction

Porcine epidemic diarrhea virus (PEDV) is an alphacoronavirus belonging to the *Coronaviridae* family and *Orthocoronavirinae* subfamily; it replicates in enterocytes of the small intestine and causes enteric disease in pigs, with clinical signs of vomiting, anorexia, diarrhea and dehydration [1]. All age groups of pigs are susceptible to porcine epidemic diarrhea; however, the highest mortality occurs in suckling piglets <10 days old and may reach 100%. In older pigs, the virus causes milder disease symptoms and lower mortality rates [2]. This coronavirus was first discovered in the UK and Belgium in the early 1970s [3]. Afterward, PEDV rapidly spread to many countries of the world, including those of Asia and North America, where it has established itself endemically and causes sporadic outbreaks of varying severity [4].

This pathogen is one of the largest RNA viruses, and like other coronaviruses, has a single-stranded, positive-sense RNA genome (excluding polyA) of about 280,000 bp with a 5′ cap and a 3′ polyadenylated tail. The genome of PEDV comprises 5′ and 3′ untranslated regions and seven open reading frames (ORFs), which are ORF1a, ORF1b and ORFs 2–6. The ORFs encode four structural proteins—the glycosylated spike protein (S), the glycosylated membrane protein (M), the envelope protein (E) and the RNA-binding nucleocapsid protein (N)—and three nonstructural polyproteins required for transcription and translation—ORF1a, ORF1b and ORF3 (Figure 1) [5].

The M protein is a part of the viral envelope which plays an important role in the assembly of the virus and the release of viral particles. The M protein can induce the generation of neutralizing anti-M antibodies in the presence of the complement and stimulate the expression of interferon alpha genes [6]. The N protein is involved in viral replication and transcription and binds to the viral RNA forming complex that serves as the core of PEDV. The E protein is vital to viral assembly and budding. Lastly, the S protein is an envelope glycoprotein with a molecular weight of approximately 180 kDa and contains the N terminal S1 region of amino acid (aa) residues 1–735 and the C terminal S2 region of aa residues 736–1383. The S1 region can be divided into five subdomains—S10, S1A, S1B, S1C and S1D—and is involved in virus–receptor recognition and binding, while the S2 region forms the transmembrane structure of the S protein and mediates cell membrane fusion. Furthermore, the whole S protein stimulates the induction of neutralizing antibodies in the host [6,7].

Despite only one PEDV serotype having been described, phylogenetic studies based on the *S* gene indicated that PEDV can be divided into two main groups, these being genogroups 1 (G1) and 2 (G2), which can be further divided into subgroups. The G1 genogroup comprises the classical strains, including the prototype PEDV strains found in the earliest research in Europe and Belgium (the CV777 strain) and cell culture-adapted vaccine strains obtained via successive in vitro passaging. The G2 genogroup includes global field isolates prevalent around the world which feature typical insertions and deletions in the *S* gene compared to the G1 PEDV strains. This may be the reason why the currently available commercial vaccines based on the attenuated CV777 strain failed to provide effective protection against epidemic G2 PEDV strains [5,8,9,10]. Strains of the G1 genogroup present low-to-moderate virulence while strains of the G2 group show higher infectivity and virulence [11].

Outbreaks of severe watery diarrhea with high morbidity and mortality in young piglets are usually the first signs indicating the circulation of PEDV in a herd. A diagnosis of PEDV infection cannot be made without molecular and/or immunological methods because the clinical signs and histopathological lesions associated with PEDV are similar to those caused by other porcine enteric coronaviruses, including porcine deltacoronavirus (PDCoV) or transmissible gastroenteritis virus (TGEV) [1]. Many methods have been developed for the detection of PEDV and have been reported in numerous studies. The objective of this paper was to summarize the currently available serological and molecular assays for the diagnosis of PEDV, to highlight their diagnostic performance and to emphasize the advantages and drawbacks of the application of individual tests.

## 2. Literature Search Strategy

Literature related to the topic was selected by searching on the PubMed and Google Scholar electronic databases. The following keywords were used to search the published articles: PEDV, ELISA, epitope, M, N, S, fluorescent microsphere immunoassay, FMIA, virus neutralization assay, VN, indirect immunofluorescence assay, IFA, immunochromatography assay, IC, lateral flow, RT-PCR, real-time PCR, qRT-PCR, loop-mediated isothermal amplification assays, LAMP, CRISPR, NGS, TaqMan, Sybr Green and sequence. The keywords were used in combinations.

## 3. Serological Methods for PEDV Detection

### 3.1. Enzyme-Linked Immunosorbent Assay for the Detection of Antibodies against PEDV

Several commercial and in-house PEDV enzyme-linked immunosorbent assays (ELISAs) have been developed for the detection of immunoglobulin G or mucosal immunoglobulin A in serum, milk, colostrum samples, oral fluid and meat juice. Two variants of such assays are used for the detection of anti-PEDV antibodies: indirect ELISA and competitive or blocking ELISA. In the indirect assay, the antigen is coated onto the surface of the microplate well and incubated with serum samples to facilitate the formation of an antibody–antigen complex. The primary antibody is then allowed to react with the enzyme-labeled secondary antibody; following this, color develops (Figure 2).

Early indirect ELISAs were based on whole-virus preparations derived from Vero cells [12]. The advantage of assays using whole viral antigens is that whole-virus proteins are expressed, which increases the sensitivity of such tests and simplifies their production. On the other hand, such ELISAs may detect nonspecific background signals and cross-react with other viral antibodies, which consequently reduces their specificity [12]. More recently, indirect ELISAs have used recombinantly expressed and purified structural proteins of PEDV. The antigens used in most of the ELISAs developed so far have been expressed using prokaryotic vectors for protein expression. Although this system achieves high protein yields, it does not offer proper protein folding, which may reduce the diagnostic sensitivity and specificity of ELISAs which depend on such proteins. The precise folding and complex and appropriate post-translational modification of antigens are very important for serological assay development. Therefore, the mammalian protein expression system has recently been used for producing PEDV proteins with proper conformational structures and activity [13,14,15]. In blocking and competitive ELISAs for PEDV detection, antibodies present in the serum compete with specific polyclonal or monoclonal antibodies, blocking their binding to antigens immobilized in the wells. In these assays, the signal is thus inversely proportional to the number of antibodies in the sample (Figure 2). These ELISAs have been confirmed to be more specific than indirect assays [13,16,17,18].

ELISAs for the detection of antibodies against PEDV are mainly based on the major structural proteins (S, M, N and E) as antigens. The commercially available ELISAs predominantly use the nucleocapsid protein because it is the most abundantly expressed protein and induces a strong humoral response. Additionally, the N protein is highly conserved in different PEDV strains and is produced most intensively during the early stages of infection, rendering it the most appropriate antigen for early PEDV diagnosis [19,20]. The expression of the N protein in the *E. coli* expression system was also discovered to be higher than the expression of the S protein [21]. However, some studies revealed that the N and M proteins, as well as the whole virus, cross-reacted with other porcine coronaviruses such as TGEV and porcine respiratory coronavirus [22,23]. To solve this issue, some researchers have developed immunoassays with truncated PEDV N and M proteins [23]. The epitopes within the PEDV N protein have not been well studied or characterized, although it is known that the N protein does not contain any neutralizing epitopes. To date, only two epitopes (NEP-D4 and NEP-D6) on the N protein have been identified and characterized. Using these antigens in assays, no cross-reaction between PEDV and TGEV was observed, suggesting that they could be utilized to develop precise diagnostic assays for the detection of PEDV [20].

The immunogenicity of the S protein is markedly higher than that of the N protein, and thus, spike-based ELISAs are more sensitive and specific than nucleocapsid-based ones [13,14,16,21,22,24,25]. Moreover, antibodies against the S protein are detectable in serum samples over a longer period than antibodies against the N protein. In contrast to the N protein, the S protein contains multiple neutralizing epitopes capable of inducing neutralizing antibody production; therefore, it is a good candidate for the detection of such antibodies [13]. The spike protein is considered to be the most antigenic PEDV protein. Many neutralizing epitopes on the PEDV S protein have been identified, including COE (aa residues 499–638) [26], S1D (aa residues 636–789) [27] and 2C10 (aa residues 1368–1374) [28]. The S10, S1A and S1B subregions and the 1368GPRLQPY1374 peptide motif found on the carboxy terminal of the spike protein were also suggested to have neutralizing epitopes [14,28,29]. Most recently, additional B cell epitopes, namely 592TSLLASACTIDLFGYP607 [30], SE16 (722SSTFNSTREL731) [31], SS2 (748YSNIGVCK755) and SS6 (764LQDGQVKI771) [30], have been identified as linear epitopes which could be used for developing methods to detect PEDV. Most of the neutralizing epitopes are located in the S1 region; therefore, this region is the main target for the construction of unique PEDV diagnostic methods [7,14]. However, the high molecular weight of the full-length S protein containing both S1 and S2 epitopes presents a major technical challenge for large-scale recombinant protein expression and production. Therefore, many researchers have preferred to use only the S1 protein instead of the full-length S glycoprotein [14,15,32]. Moreover, as was shown by Chang et al., an S1-based ELISA had even higher sensitivity and specificity than an ELISA based on the full-length S protein [15]. The only limitation to the use of this subregion is that it has a particularly high degree of genetic diversity compared to the full-length S protein, which itself is diverse. Therefore, using S1-based ELISA tests, an antibody response may not always be detectable against all circulating PEDV strains.

The PEDV M protein is a structural membrane glycoprotein localized to the virus surface and is the most abundant component of the viral envelope. Thus far, one linear B-cell epitope, ^195^WAFYVR^200^, has been screened by hybridoma technology. As was shown, this epitope (designated M-14) was conserved across different PEDV strains and did not cross-react with sera positive for TGEV; thus, it could be used to differentiate PEDV-positive sera from TGEV-positive sera [33]. However, achieving the expression of the full-length M protein is complex and challenging due to many unknown factors, including its sprout function, which can lead to the damage of cell wall structures. The low level of expression of the M protein which is possible limits its potential for use as a diagnostic antigen for application in ELISAs [33]. The E protein is also a diagnostic marker, and this characteristic will be helpful in the development of novel serological assays, as well as in the design of vaccines. The study performed by Lei et al. revealed that besides S1, the recombinant ORF3C and E proteins may also be used as antigens for the detection of anti-PEDV antibodies; however, the S1 protein demonstrated the highest sensitivity [34]. The reactivity of nonstructural protein 1 (Nsp1), Nsp2, the ADP-ribose-1”-monophosphatase domain of Nsp3 and the acidic domain of Nsp3 was less pronounced, indicating that they should be excluded from consideration as novel diagnostic antigens [34].

All the aforementioned methods utilize conventional PEDV-specific polyclonal or monoclonal antibodies which exhibit high levels of instability. This may reduce production yields, and a further disadvantage of these antibodies is their high cost [35]. Single-domain antibodies (sdAbs), called nanobodies, are the smallest antibodies possessing antigen-binding activity and they have been used to overcome these problems. Due to the lack of light chains, nanobodies exhibit favorable features which traditional antibodies lack, such as thermal stability, permeability, high binding affinity and ease of production in prokaryotic and mammalian expression systems. Recently, biotinylated nanobodies against the PEDV N protein (Nb2) have been used to develop a blocking ELISA for the detection of antibodies against PEDV. This newly developed test showed 93.18% specificity and 100% sensitivity [36]. Another indication of nanobody suitability came in the use of a truncated PEDV S protein spanning aa residues from 444 to 770 and containing most neutralizing epitopes, which was used as an antigen to select S protein-specific sdAbs. The results suggested a potential application of the single-domain antibody named S7, the specificity of which has been confirmed using ELISA and immunocytochemistry. However, S7 did not neutralize PEDV at all, suggesting that this antibody is not suitable for vaccine development [37]. Although the use of nanobodies has many advantages, their generation is quite complicated because it requires the screening of functional antibodies via the immunization of camels, the construction of libraries and phage display [36].

As an example of recent innovation, an amplified luminescent proximity homogenous assay (AlphaLISA) for the detection of antibodies against PEDV has been developed. This bead-based assay platform, originally developed by PerkinElmer, operates via donor and acceptor beads being coupled to antibodies or proteins which interact with the target analyte, bringing the beads into the proximity of each other and leading to energy transfer and the emission of a chemiluminescent signal. The AlphaLISA method allows the detection and characterization of pathogen-specific antibodies with greater speed, sensitivity and simplicity of use [38]. Additionally, an ELISA-like multiplex planar immunoassay (the AgroDiag PorCoV) has been developed, which has PEDV-specific recombinant S1 proteins printed in an array of spots at the bottom of a microplate for the simultaneous detection and differential diagnosis of PEDV, TGEV and PDCoV in a single sample. The technology and working principle are similar to those of the solid-phase standard ELISA. The reaction is visualized as blue spots, with intensity correlating with the levels of antibodies specific to the viral antigen target in the array. The overall diagnostic sensitivity was 92% for PEDV, 100% for TGEV and 98% for PDCov, while the diagnostic and analytical specificity for each antigen target was 100%, demonstrating that this assay is an efficient and reliable test for the differential detection and serodiagnosis of PEDV, TGEV and PDCoV [39].

### 3.2. Fluorescent Microsphere Immunoassay

The fluorescent microsphere immunoassay (FMIA) is a relatively new serodiagnostic tool for the sensitive, specific, rapid and simultaneous detection of antibodies against multiple pathogens. The FMIA utilizes multiple color-coded beads (fluorescent microspheres) whereby each bead can be conjugated to a different antigen that binds to an antibody in biological samples [40]. This assay is based upon xMAP technology, which uses different bead sets internally dyed with different fluorescent preparations. In an FMIA, similarly to capture ELISAs, a capture antigen is covalently attached to a bead surface and binds to a target antibody present in the sample. Antigen-conjugated microsphere–antibody complexes are analyzed using dual-laser instruments and the results are shown as the median fluorescent intensity (Figure 3). Compared to other methods, the FMIA has many advantages. It is more sensitive and specific than an ELISA and can simultaneously detect antibodies against multiple pathogens (up to 500) in a single tested sample. Beads with immobilized PEDV-specific antigens can be used with beads specific for other swine pathogens, including swine influenza virus, porcine reproductive and respiratory syndrome virus, TGEV and other pathogens, for the detection of antibodies against all of them in one assay. The fluorescent microsphere immunoassay is a rapid multiplexing platform and can therefore be applied in large-scale testing. Additionally, FMIA is less labor-intensive than most other methods and requires only a small sample volume for analysis, which is important when the number of clinical samples is limited [4]. Despite these merits, the FMIA is a rarely used method to detect PEDV infection, notwithstanding that in 2015, an FMIA based on the nucleocapsid protein of PEDV was developed for the detection of PEDV antibodies in serum samples. The specificity and sensitivity of this test were 99.2% and 98.2%, respectively, and the results yielded by this assay correlated strongly with those given by ELISAs and indirect immunofluorescence assays (kappa scores >0.91) [16].

### 3.3. Virus Neutralization and Indirect Immunofluorescence Assays

The virus neutralization assay (VN) was previously the most widely employed serological assay for the detection of anti-PEDV-specific antibodies because of its high specificity, which makes the detection of PEDV-neutralizing antibodies possible as early as 7 days post-infection (dpi) [41]. Such antibodies block viral replication, neutralizing or reducing virus infectivity. Originally, the assay which was used to detect neutralizing antibodies against PEDV was a VN assay based on the cytopathic effect (CPE) [24]. However, a VN assay is time-consuming because of the biological processes involved, namely that the test needs to wait until a viral CPE is fully developed; it is also expensive, and it requires well-trained technicians since the direct observation of CPE in cell cultures is subjective. Furthermore, the interpretation of results may be complicated because of cytotoxic effects, and misinterpretations may occur, especially at lower serum dilutions. Therefore, the PEDV fluorescent focus neutralization (FFN) assay has been developed for the rapid detection of PEDV-neutralizing antibodies [16]. The FFN assay is a modification of the VN assay in which the cytopathic effect is not interpreted; thus, this assay has a shorter turnaround time [16]. In this assay, plates with Vero cell cultures inoculated with a virus–serum mixture are incubated for 24–48 h, and then, stained with a fluorophore-conjugated antibody against PEDV. The PEDV-specific antibody level in serum can be estimated based on fluorescence intensity. In addition to measuring neutralizing antibody levels in serum, the FFN assay has also been optimized to quantify PEDV-neutralizing antibodies in colostrum and milk to monitor the lactogenic immunity of suckling piglets [16,42]. Besides VNs and FFNs, imaging cytometry is also an appropriate method for the detection of neutralizing antibodies. Imaging cytometry is utilized in the high-throughput virus reduction neutralization test (HTNT) which was developed by Sarmento et al. [43]. This test offers a more objective and semi-automatic approach that excludes human subjectivity from the examination process and shortens the reading time for a 96-well plate to less than 4 min. The HTNT showed excellent sensitivity and specificity, confirming its value as a tool for the detection of neutralizing antibodies [43].

The indirect immunofluorescence assay (IFA) has been commonly used to identify anti-PEDV antibodies in serum samples and to check the immune status of herds to PEDV [41,42,44,45]. In this method, Vero cell cultures infected with PEDV are usually used as the antigen. Then, the serum samples to be tested are added to the plate. If antibodies against PEDV are present in the serum, they bind to the virus antigens attached to the plate wells; then these antibodies may be detected by adding fluorophore-conjugated anti-swine secondary antibodies and inspecting the wells using a fluorescence microscope [4,41,42]. The IFA has specificity comparable to an FFN assay but lower diagnostic sensitivity. The time required for an IFA is shorter than for an FFN assay and it is easier to implement; however, the results of an IFA do not correspond to the neutralizing antibody response displayed by the FFN test since the IFA detects IgG antibodies [4,5]. Similarly to the VN assay, the IFA is not automated and is subjective with respect to result reading. Moreover, it was shown that while anti-PEDV antibodies are detectable by the IFA assay in the 1–2 weeks following PEDV exposure, the antibody titers drop to levels undetectable in this assay earlier than they do in the FFN assay [4].

### 3.4. Immunochromatography Assay

An immunochromatography (IC) assay, also known as a lateral flow test, is a point-of-care device which provides qualitative or semi-quantitative information. The IC is a simple-to-use and rapid immunoassay, the result of which can be read within several minutes without professional equipment or qualified personnel. The IC assay is also very stable and is relatively inexpensive to produce. Hence, this method is currently widely used as a screening test to monitor PEDV infection in field conditions [46].

The two most common formats of the lateral flow IC assay are the double-antibody sandwich assay and the competitive assay. The competitive assay is most suitable for the detection of low-molecular-weight analytes such as protein antigens, while the double-antibody sandwich assay is more suited to target analytes such bacterial pathogens and viruses with higher molecular weight [47,48]. Therefore, the IC assay developed for the detection of PEDV antigens in porcine feces is in the double-antibody sandwich format [48,49]. Common labels for lateral IC assays are carbon, liposomes, latex, colloidal carbon and gold nanoparticles, also known as colloidal gold. Depending on their physicochemical characteristics, some labels can generate a direct (visual) signal, while others produce an analytical signal, to read which additional instrumentation is required. Gold nanoparticles are currently the most widely used label in IC assays because they are inexpensive and easy to prepare, have intense color, and can be perceived by the naked eye. However, an IC assay for the detection of a PEDV N protein which utilized colloidal gold was ten times less sensitive than a real-time RT-PCR (rRT-PCR) assay [48]. The sensitivity of the RT-PCR was nevertheless surpassed by a europium (Eu) (III) chelate-based fluorescent IC assay. This assay required an immunofluorescent analyzer to obtain results; therefore, using this assay for quantitative estimation of the PEDV N protein is possible [49,50]. Moreover, a smartphone camera was also used to quantify the results of IC for detecting of PEDV [51]. More recently, a new paper-based lateral flow immunoassay utilizing color-rich latex beads as the label has been developed for the detection of PEDV in swine fecal samples [52]. This assay had high sensitivity of 88.57% and a limit of detection of 103.60 fifty-percent tissue culture infective doses/mL, and cross-reactions with other related swine viruses were not observed, signifying 100% specificity. Furthermore, this assay gave good agreement with RT-PCR results (92.59%) and was more accurate than previously reported fluorescent and colloidal gold–based IC assays [52].

A colloidal gold–based immunochromatographic strip test for the detection of PEDV-specific secretory immunoglobulin A in milk and colostrum has been developed [53]. Secretory immunoglobulin A is locally produced by plasma cells in the intestinal lamina propria and serves as the first line of immune defense in the animal gut. Therefore, this immunochromatographic test strip provides a simple method for monitoring passive lactogenic immunity in piglets [53].

The IC assay generates a result much faster than rRT-PCR, and thus, is a valuable tool for the rapid diagnosis of PEDV infection in pigs. The disadvantage of the test is that the interpretation of weak bands in the results may sometimes be problematic because the results are generally read with the naked eye. Therefore, it is important to minimize nonspecific bands in the test line. If the density of antibodies is too high, it can lead to nonspecific binding and a lack of specificity. Consequently, each test must be validated regarding the optimal density of antibodies which will minimize the occurrence of nonspecific bands. An additional possible drawback is that most IC assays are not quantitative and are not suitable for large-scale screening [48].

## 4. Methods for the Detection of PEDV Genome and/or Antigens

### 4.1. Antigen ELISA

Many quick and highly specific assays have been established for the detection of PEDV antibodies, but these antibodies can only be detected between 6 and 14 dpi, which may delay the diagnosis of PEDV [16]. Serological assays showing the presence of antibodies against PEDV are used to determine whether a pig has previously been exposed to the virus, and these assays can be used to monitor the efficacy of vaccination strategies. However, more actionable knowledge of a given epizootiological situation which is dynamic proceeds from the results of attempts to detect a virus genome that is currently infecting pigs. Fecal shedding of PEDV occurs as soon as 1–3 dpi, and PEDV RNA may be detected in fecal samples after 30 dpi [41]. Sandwich or capture ELISA assays for the earliest detection of individual PEDV antigens in fecal samples have been established [35,54,55,56,57,58]. The adoption of the sandwich format overcomes most issues associated with cross-reactivity as it employs two antibodies, each specific to a different epitope region. In such ELISAs, the wells are coated with capture antibodies specific to the S, M or N PEDV proteins, and after incubation with the sample, virus antigens bind to the capture antibody. The reaction is developed by adding an enzyme-conjugated secondary antibody which reacts with an appropriate substrate. Coloration signifies a positive result, while a lack of coloration indicates a negative one. The use of fecal samples has some advantages over the use of blood samples, including easy and non-invasive collection. On the other hand, the drawbacks include low virus concentrations and occurrences of nonspecific reactions that may reduce the sensitivity and specificity of serological assays. Additionally, fecal samples do not have constant usefulness over the course of porcine epidemic diarrhea; fecal shedding was consistently detected during the acute phase of PEDV infection but less frequently detected during the incubation and recovery periods. Therefore, the samples should be collected immediately after the appearance of clinical symptoms [55].

### 4.2. RT-PCR and qPCR

PCR constitutes one of the greatest advances in molecular biology. A practical advantage of PCR-based methods is the possibility to test different biological samples, such as rectal swabs, feces, oral fluid and intestinal samples, maintaining reasonable sensitivity and specificity across them all [4,59,60,61]. The chain reaction utilizes multiple stepwise temperature cycles and a thermostable DNA polymerase to amplify a specific DNA fragment; therefore, for the detection of viral RNA, a preliminary step is required in which a reverse transcriptase enzyme is used to convert RNA to cDNA. Several one-step or nested reverse-transcription PCR assays have been developed for the detection of PEDV and are described in the literature [19,62,63,64,65,66,67,68,69] (Table 1).

Multiplex RT-PCR makes amplification possible of more than one target fragment using more than one specific pair of primers in one reaction tube. Multiplex RT-PCRs are commonly used by veterinary diagnostic laboratories for the differential detection of related multiple pathogens, especially those causing diseases with similar symptoms. Multiplex RT-PCRs have been developed for the differentiation of PEDV from other porcine viruses [70,71,72,73,74,75,76,77] (Table 1). In standard RT-PCRs such as these, the DNA products are visualized by electrophoresis on agarose gel to reveal the presence of expected DNA bands. Using this technique, only semi-quantitative results can be achieved. Furthermore, this method can fail when samples contain low concentrations of DNA.

**Table 2 pathogens-11-01074-t002:** Summary of qRT-PCR assays for PEDV diagnosis.

PCR Type	Primer Name	Primer Sequences (5′-3′)	Target Region (Size)	Limit of Detection	Ref.
SYBR™ Green one-step qRT-PCR	mPEDNF PEDV-R	CGCAAAGACTGAACCCACTAA TTGCCTCTGTTGTTACTTGGAGAT	N (191 bp)	0.5 × 10^0.01^ TCID^50^/mL for the spiked feces and 10^0.01^ TCID_50_/mL for spiked jejunum	[78] [79]
Duplex qRT-PCR	PEDV S1-F PEDV S1-R Virulent PEDV S1-P Variant PEDV S1-P	AGGCGGTTCTTTTCAAAATTTAATG GAAATGCCAATCTCAAAGCC 5Cy5/TATTGGTGAAAACCAGGGTGTCAAT/3BHQ 2 56-FAM/TGGTTATCTACCTAGTATGAACTCCTCTAGC/3IABkFQ	S1 (191 bp for virulent PEDV; 179 bp for variant PEDV)	1 DNA copy/µL	[80]
	PEDV M-F PEDV M-R PEDV M-P	CATGGGCTAGCTTTCAGGTC CGGCCCATCACAGAAGTAGT 56-FAM/CATTCTTGGTGGTCT TTCAATCCTGA/ZEN 3IABkFQ	M (181 bp for both virulent and variant PEDVs)	1 DNA copy/µL	
Multiplex qRT-PCR (PEDV/TGEV)	PEDV-F PEDV-R PED-Cy5- P	CGCAAAGACTGAACCCACTAATTT TTGCCTCTGTTGTTACTTGGAGAT Cy5-TGTTGCCATTGCCACGACTCCTGC-BHQ3	N (198 bp)	1 TCID_50_/mL	[79]
	TGEV-F TGEV-R TGE-FAM- P	GCAGGTAAAGGTGATGTGACAA ACATTCAGCCAGTTGTGGGTAA 6FAM-TGGCACTGCTGGGATTGGCAACGA-BHQ1	N (120bp)	1 TCID_50_/mL	
qRT-PCR	PEDV S-F PEDV S-R PEDV S- P	ACGTCCCTTTACTTTCAATTCACA TATACTTGGTACACACATCCAGAGTCA FAM-TGAGTTGATTACTGGCACGCCTAAACCAC-BHQ	S (111 bp)	10^−0.2^ TCID_50_/mL	[81]
	PEDV N-F PEDV N-R PEDV N-P	GAATTCCCAAGGGCGAAAAT TTTTCGACAAATTCCGCATCT FAM-CGTAGCAGCTTGCTTCGGACCCA-BHQ	N (87 bp)	10^−2.2^ TCID_50_/mL	
Multiplex qRT-PCR	F-C R-C C-P	GTCGTTGTTTTGGGTGGTTA CCATGAACGCCACTATCAGT FAM-TAGCTGGTACTGTGGCACAGGCATTG-BHQ1	S (89 bp for classical PEDV)	5 × 10^2^ DNA copies/reaction	[67]
	F-V R-V V-P	GTTGTACTGGGCGGTTATCT CCATGAACGCCACTAGCAGT VIC-TGGTACTGTGCTGGCCAACATCCA-BHQ1	S (98 bp for variant PEDV)	5 × 10^2^ DNA copies/reaction	
Duplex qRT-PCR (PEDV/PDCoV)	PEDV rF PEDV rR PEDV rP	GGTTGTGGCGCAGGACA CGGCCCATCACAGAAGTAGT FAM-CATTCTTGG/ZEN/TGGTCTTTCAATCCTGA-IABkFQ	M (79 bp)	7 RNA copies/reaction	[82]
	PDCoV rF PDCoV rR PDCoV rP	TGAGAGTAGACTCCTTGCAGGGA GAGAATTGGAGCCATGTGGT NED-TGTACCCATTGGATCCATAA-MGB	M (105 bp)	7 RNA copies/reaction	
Multiplex qRT-PCR (PEDV/PDCoV/SADS-CoV/PToV)	PEDV-F PEDV-R PEDV-P	CTCCCTTGAATTTGAGTTCG ACCACCTGTAACCTTGATAC FAM-TTACCAACAGCCTTATTAAGCAC-MGB	ORF1a (85 bp)	1 × 10^2^ copies/µL	[83]
	PDCoV-F PDCoV-R PDCoV-P	AAAGCTTTCAAGACAATACCT TACGACAAACTCCTGAAAGCA Texas Red-TACGATACGACTGCATTGGCCTAC-BHQ2	ORF1b (87 bp)	1 × 10^2^ copies/µL	
	PToV-F PToV-R PToV-P	TCATCCACCCAGTTCAAAT TGCACAATTCTCTCTCCAAAT VIC-CCTCAGaTTTCGaAGATAGaACC-BHQ1	ORF1a (73 bp)	1 × 10^2^ copies/µL	
	SADS-CoV-F SADS-CoV-R SADS-CoV-P	CATTTGCCGTTCTTGACCAT AACCCAGCAATTGTTATCTGAA Cy5-CAAGTGCACGCTTACCATCAACTACT-BHQ3	ORF1a (95 bp)	1 × 10^2^ copies/µL	
5-Plex qRT-PCR (PEDV/PDCoV/TGEV/SADS-CoV)	PEDV-N1195-F PEDV-N1269-R PEDV-N1221-P	GAAGAGGCCATCTACGATGATGT AACAGCTGTGTCCCATTCCAA JUN/TGTGCCATCTGATGTGACTCATGCCA/QSY	N (75 bp)	8 genomic copies/reaction	[84]
	PDCoV-N-F2 PDCoV-N-R2 PDCoV-N-P	CCAGACATGTGCCTGGTGTT CCCYGCCTGAAAGTTGCT ABY/ARATGCTTTTCGCTGGCCACCTTG/QSY	N (68 bp)	4 genomic copies/reaction	
	TGEV-S-F2 TGEV-S-R2 TGEV-S-P	GTGGTAATATGYTRTATGGCYTACAA GCCAGACCATTGATTTTCAAAACT VIC/TTGCTTATTTACATGGTGCYAGT/MGB	S (101 bp)	16 genomic copies/reaction	
	SADS-N-F3 SADS-N-R3 SADS-N-P	CCAGGCCTCAAAGTGGTAAAAA TGCTTACGAGCCGGTTTAGG FAM/ACCCAAACC/ZEN/AAGAAGCAGAGCTGTCTCAC/QSY	N (85 bp)	6.8 genomic copies/reaction	
Multiplex qRT-PCR (TGEV/PEDV/PDCoV/PEAV)	PEDV-F PEDV-R PEDV-P	GATACTTTGGCCTCTTGTGT CACAACCGAATGCTATTGACG FAM-TTCAGCATCCTTATGGCTTGCATC-TAMRA	M (150 bp)	100 copies/reaction	[85]
	PDCoV-F PDCoV-R PDCoV-P	ATTTGGACCGCAGTTGACA GCCCAGGATATAAAGGTCAG Cy5-TAAGAAGGACGCAGTTTTCATTGTG-BHQ2	M (92 bp)	100 copies/reaction	
	TGEV-F TGEV-R TGEV-P	TGCCATGAACAAACCAAC GGCACTTTACCATCGAAT HEX-TAGCACCACGACTACCAAGC-BHQ1	N (81 bp)	100 copies/reaction	
	PEAV-F PEAV-R PEAV-P	TCTCGGCTTACTCTAAACCC CATCCACCATCTCAACCTC TexasRed-AAGACCTAAATGCTGATGCCCCA-BHQ2	N (150 bp)	100 copies/reaction	
Duplex SYBR Green qPCR (PEDV/ PBoV 3/4/5)	PEDV F PEDV R	GAGGGTGTTTTCTGGGTTG TGCCTCTGTTGTTACTTGG	N (226 bp)	10 copies/μL	[86]
	PBoV F PBoV R	GGTGATCCTGTCAATAAA TGCAAAGAGTCGATAAAGT	VP1 (131 bp)	10 copies/μL	
TaqMan probe-based qPCR	eU-ORF3 F eU-ORF3 R FP2 F RP2 R P	GCCGAATTCATGTTTCTTGGACTTTTTCAA ACGCTCGAGTCATTCACTAATTGTAGCATA CGTTTTGCTGTCATTGTTCTT AGACTAAACAAAGCCTGCCAATA FAM-ATTGCCCACTTTTATATTATTGTGGTGCATTTTTAGATG-TAMRA	ORF3 (675 bp for virulent strains and 626 bp for vaccine strains)	37 copies/reaction	[87]
			ORF3 (106 bp for virulent strains and 57 bp for vaccine strains)	37 copies/reaction	
SYBR Green I-based duplex qRT-PCR (PEDV/PCV3)	PEDV-F PEDV-R	AAATGGGAAGTCGGCAGA GTTTTGTTGTGGCGGTAG	ORF1 (163 bp)	3.46 × 10^1^ copies/μL	[88]
	PCV3-F PCV3-R	CTACGAGTGTCCTGAAGA CCTCCACACTCCACAATA	Rep (136 bp)	6.12 × 10^1^ copies/μL	
Multiplex EvaGreen qPCR (TGEV/RVA/RVC/PEDV/PCV2)	PEDV- F PEDV- R	GGCGGATACTGGAATGAGCAA CGGTCGGCGTGAGGTCCTGTT	N (110 bp)	5 copies/μL	[89]
	TGEV-F TGEV-R	ATGGTGTTAGGTGATTATTTTCC AATACAATGCTTTAAGATTTTCCA	S (106 bp)	5 copies/μL	
	RVA-F RVA-R	TGAAGTGAGGACCAGGCTAA ACGAAATCACACCCTTACTTG	VP6 (97 bp)	5 copies/μL	
	RVC-F RVC-R	TGTTGCATCCGTGAAGAGAATGGT GCATTAGCCCCTACGCAAGC	VP6 (126 bp)	5 copies/μL	
	PCV2-F PCV2-R	ATCCGAAGGTGCGGGAGA TGACGTATCCAAGGAGGCG	CP (162 bp)	50 copies/μL	
Duplex qRT-PCR	F-C/V R-C/V C-Probe (classical PEDV) V-Probe (variant PEDV)	GCTAGTGGCGTTCATGG TGTAAATAAAGCTGGTAACCAC CY5-TACATCGATTCTGGTCAGGGCTTTGAG-BHQ2 FAM-CCATATTAGAGGTGGTCATGGCTTTGAG-BHQ1	S1 (110 bp)	4.8 × 10^2^ DNA copies/reaction	[68]
qRT-PCR	PEDV F PEDV R PEDV P	CAGGACACATTCTTGGTGGTCTT CAAGCAATGTACCACTAAGGAGTGT FAM-ACGCGCTTCTCACTAC-MGB	M (240 bp)	10 copies/ml	[69]

F—forward; R—reverse; P—probe; TGEV—transmissible gastroenteritis virus; GAR—porcine group A rotavirus; PDCoV—porcine deltacoronavirus; SADS-CoV—swine acute diarrhea syndrome coronavirus; PToV—porcine torovirus; PEAV—porcine enteric alphacoronavirus; PBoV—porcine bocavirus; PCV3—porcine circovirus 3; RVA—rotavirus A; RVC—porcine rotavirus C; PCV2—porcine circovirus 2; Ref.—reference; bp- base pair.

qRT-PCR assays are currently favored because they are more sensitive, faster and much easier to perform than standard RT-PCRs and can provide quantitative detection. Many qRT-PCR test protocols have been published by researchers around the world for the specific detection of PEDV after satisfactory evaluation of their sensitivity and specificity using clinical samples or samples originating from experimentally infected pigs [67,68,69,78,89] (Table 2). Some commercial multiplex qRT-PCR assays for the simultaneous detection and differentiation of PDCoV, PEDV and TGEV have been developed (e.g., the VetMAX PEDV/TGEV/SDCoV (PDCoV) Kit from Applied Biosystems). The major advantages of qRT-PCR methods are high sensitivity, throughput and process automation, the quantification of viral loads and the possibility of the simultaneous (multiplexed) identification and discrimination of different swine pathogens. Two real-time PCR methods are the most popular: TaqMan and SYBR Green. The SYBR Green method uses a fluorescent nonspecific DNA-binding dye, while the TaqMan method uses a specially designed probe that is complementary to the internal fragment of the target DNA. Both of them have high sensitivity. The TaqMan-probe method has higher specificity but it is less versatile in its application since one probe is able to detect only one specific fragment of DNA. Although multiplex real-time PCR is a powerful genetic method, the development of an application of this method is sometimes challenging. The presence of multiple primer pairs in multiplex PCR assays may result in the formation of nonspecific products. Therefore, special attention should be paid to designing specific primers and probes [84]. The primer length, melting temperature, different annealing temperatures for different primers, guanine–cytosine content, secondary structure, repeats, 3′ end stability, product position and optimization of PCR conditions have to be considered. In particular, the primer pairs should be not complementary to each other. In order to block nonspecific priming, a novel dual-priming oligonucleotide (DPO) system was recently used in multiplex RT-PCR assays for the specific detection of PEDV and other swine viruses, including TGEV, PRV-A, PDCoV and swine acute diarrhea syndrome coronavirus (SADS-CoV) [90,91]. The DPO primers have a special structure which is different from that of conventional primers. They contain two separate sequence fragments linked through a polydeoxyinosine linker, which prevents the formation of primer-dimer and hairpin structures and therefore eliminates nonspecific interactions. Moreover, using the DPO primers, optimization of the annealing temperature is not required since annealing temperatures already differ for the two primer fragments as a function of their structure. This simplifies primer design [90,91]. Because coronaviruses evolve rapidly, the selection of the most suitable pair of primers is a matter of critical importance. The selection process should attempt to specify primers that will be able to detect all circulating variant PEDV strains and that will be able distinguish PEDV from other closely related viruses, such as PDCoV or TGEV. Updated PEDV sequence information is required at the time of the experiment design to ascertain whether the primers and probes designed for the virus sequences available at the time of their formulation are suitable for detection of the new virus strains.

### 4.3. Loop-Mediated Isothermal Amplification Assay

Recently, a variety of isothermal nucleic acid amplification techniques have been developed to improve conventional PCR methods. Insulated isothermal amplification systems do not offer the ability to multiplex, but might provide the opportunity to deploy tests in the field in countries where laboratories are not easily accessible. These methods include loop-mediated isothermal amplification (LAMP) assays, strand displacement amplification [92], recombinase polymerase amplification (RPA) [93], and cross-priming amplification [94]. A method among these which offers a simple, effective and rapid alternative to classical or real-time PCRs is LAMP. This type of amplification uses four to six primers that recognize six to eight DNA target regions in association with Bst polymerase, which has strand-displacement activity [95]. The method does not require expensive equipment such as a thermocycler; it only requires a water bath or heat block, and the entire procedure can be completed within 60 min. The application of LAMP tests might reduce the cost of detection of coronavirus. Another recommending factor of these assays is that the amount of amplified DNA that they produce is high compared to the standard PCR assays.

A number of LAMP-based detection methods have been developed and applied to the detection of PEDV. The amplified product of these assays can be detected as the precipitation of magnesium pyrophosphate or fluorescent dye and is visible to the naked eye, providing rapid qualitative results. Furthermore, turbidity can also be measured in real time, allowing quantitative analysis of the target DNA. However, fluorescent dye detection methods require additional instruments such as a fluorescence detector, which limits the applicability of reverse-transcription LAMP (RT-LAMP) as a field diagnostic assay [96]. Recently, hydroxynaphthol blue (HNB), a metal colorimetric indicator, was also used for the detection of PEDV [96]. The reduction in Mg2+ concentration in a LAMP solution causes a color change in the HNB solution from purple to sky blue, granting a means of direct visual detection. This method is expected to be useful for routine application in the field. The only drawback of this method may be the difficulty in distinguishing the color change between positive and negative samples when the amount of the template in the tested sample is low [96].

The first RT-LAMP assay to amplify PEDV was developed by Ren and Li and was based on the detection of a fragment of the PEDV *N* gene [95]. The detection limit of this technique was lower than those of standard RT-PCR and antigen-capture ELISA; however the sensitivity and specificity of all the tests were similar [95]. A real-time RT-LAMP method, which was developed by Yu et al. and involved loop primers for the detection of the PEDV *M* gene, had similar sensitivity to that of real-time PCR and was 100 times more sensitive than one-step RT-PCR [97]. Additionally, the sensitivity of the real-time fluorescent RT-LAMP assay based on the detection of the PEDV *M* gene and using real-time fluorescent devices was at least 100 times higher than that of one-step RT-PCR [98]. A PEDV *M* gene real-time RT-LAMP also had higher sensitivity than the RT-LAMP developed for the *N* gene [97]. Additionally, Gou et al. established a nucleic acid visualization technique that combined RT-LAMP and a vertical flow (VF) visualization strip to detect PEDV. This M-based method demonstrated high specificity for PEDV and similar sensitivity to RT-PCR and RT-LAMP [99]. An advanced visual RT-LAMP assay combined with HNB for visual detection was 1000 times more sensitive than standard RT-PCR and comparable to rRT-PCR [97]. Most recently, Zhou et al. [100] developed a rapid, simple tool for the simultaneous detection of PEDV, PDCoV and SADS-CoV, which integrated a microfluidic chip, a real-time RT-LAMP assay and a portable microfluidic chip fluorescence detector. The centrifugal microfluidic lab-on-a-chip is the most recent innovation in the microfluidic field, and it enables multiplex high-throughput research with high efficiency and speed [101,102]. This newly developed microfluidic-RT-LAMP chip detection system presented good stability (coefficient of variation (C.V.) < 5%), a specificity of 100% and sensitivity levels of 92.24%, 92.19% and 91.23% for PEDV, PDCoV and SADS-CoV, respectively [100].

### 4.4. Sequencing

The genetic diversity of PEDV strains may pose a fundamental problem for PEDV testing and diagnosis. Nucleotide sequencing analyses are useful tools to identify and monitor the genetic variation and evolution of circulating PEDV strains and determine recombination events. Comparing the nucleotide and amino acid sequences of local or regional circulating or reemerging PEDV strains with those used for the development of immunological and molecular assays ensures the reliability of those tests for as long as the regional strains and the assay development strains have high identity. Therefore, sequence information for the PEDV field strains circulating in each country should be constantly updated.

The sequences of several PEDV genes have been previously analyzed, including the *N*, *S*, *M* and *ORF3* genes. Those of the *S* gene are mainly used as genetic markers for determining the genotypes of PEDV strains [103,104,105,106,107,108,109,110]. Sanger sequencing and next-generation sequencing (NGS) are the two main sequencing methods currently used for determining the PEDV sequence, with Sanger sequencing being the traditional sequencing method that needs the design and synthesis of virus-specific primers and is only capable of the sequencing of one DNA fragment at a time. In the Sanger-sequencing approach, DNA primers designed for this particular purpose are annealed to the amplified target DNA (amplified product), and then, extended by DNA polymerase by incorporating DNA nucleotides (dATP, dGTP, dCTP and dTTP) complementary to the DNA template. In addition, small amounts of fluorophore-labeled chain-terminating dideoxynucleotides (ddATP, ddGTP, ddCTP and ddTTP) are randomly incorporated for each nucleotide. The output of Sanger sequencing is a four-color chromatogram representing the fluorescence peak intensities associated with the ddNTPs (each labeled with a different fluorescent dye) along the DNA sequence. Sanger sequencing can read up to 500–1000 bp per reaction without complicated data analysis [111]. While Sanger sequencing is currently extensively and widely used, NGS technology is being adopted increasingly often for whole-genome sequencing. In NGS, genomic material in a clinical specimen is fragmented, randomly amplified and used to prepare a library of genomic fragments that are then sequenced by using one of several common strategies. A major advantage of NGS is that it has no need for target-specific primers, unlike Sanger sequencing. By virtue of this, NGS has the ability to discover novel, unknown PEDV strains and provides the opportunity to detect recombination events in PEDV strains [112,113]. It also sequences a full-length viral genome rapidly in one sequencing run. Although NGS sequencing technology is the foundation of valuable techniques for the diagnosis of entire genomes of viruses, it has some disadvantages, including cost and the requirements for specialized instruments and trained personnel. In particular, proficiency in using advanced bioinformatic tools is required for assembly and mapping to reference genomes. Furthermore, the detection of viral genomes can be problematic because of the low amounts of genetic material sometimes contained in clinical samples. Therefore, the applicability of NGS in clinical diagnostic methodologies is very limited.

### 4.5. CRISPR–Cas Technology

Clustered regularly interspaced short palindromic repeats (CRISPR)–Cas is a nucleic acid detection technique which allows specific sequences in the genome to be added, removed or modified using the Cas12 and 13 proteins. By combining Cas proteins with a preamplification step such as PCR, RPA or LAMP and a single-stranded DNA–fluorescently quenched (ssDNA-FQ) reporter, various biosensing platforms have been developed. This system shows great potential for integration into novel, accurate, fast and convenient molecular diagnostic methods [114].

Yang et al. first created a rapid method to distinguish PEDV wild-type strains from attenuated vaccine strains by combining reverse-transcription–enzymatic recombinase amplification (RT-ERA) with the CRISPR–Cas12a system [115]. The total detection time of this assay was approximately 30 min and the results were visible under blue LED light. The method was sensitive, being capable of detecting two copies of genomic DNA, and specific, showing no cross-reactivity with other porcine viruses. Recently, Liu et al. developed a multiplex nucleic acid detection method based on CRISPR–Cas12a and multiplex RT-LAMP for the detection of four porcine diarrhea coronaviruses: PEDV, TGEV, PDCoV and SADS-CoV. This newly developed assay achieved single-copy sensitivity with no cross-reactivity. The results were visible to the naked eye using a ROX-labeled ssDNA-FQ reporter [116]. Generally, the methods described above are sensitive, capable of rapid detection, do not require expensive equipment or long training and can be performed at the point of care.

## 5. Conclusions

In the absence of effective vaccines, early, rapid and accurate diagnosis of PEDV is crucial for the prevention and control of the spread of disease. Therefore, the development and selection of reliable, appropriate diagnostic tests is very important. Multiple approaches have been explored and methods implemented to improve the detection of PEDV. Each of the tests has its own unique advantages and disadvantages (Table 3), and thus, as far as is practicable, several methods are combined to avoid the drawbacks of using a single method. While molecular assays allow the identification of the nucleic acid of PEDV, serological assays provide information about the prevalence of PEDV infection and the efficiency of vaccination strategies, and may be helpful in predicting the duration of immunity in piglets. Molecular assays have high sensitivity and specificity but require trained staff and specialized equipment, and the associated costs can be high. In contrast, antibody-based assays are generally cheaper and generally more accessible to untrained users. However, serological techniques are less sensitive than nucleic acid detection methods. While molecular and antibody-based methods will likely continue to dominate in the diagnosis of PEDV, newer technologies such as NGS could complement them and enhance their utility. Next-generation sequencing may provide valuable information on PEDV heterogeneity and evolution and may be applied to identify new or reemerging viruses. Furthermore, the new generation method based on nonspecific DNA cleavage in CRISPR–Cas systems promises major advances in the diagnosis of PEDV.

## Figures and Tables

**Figure 1 pathogens-11-01074-f001:**
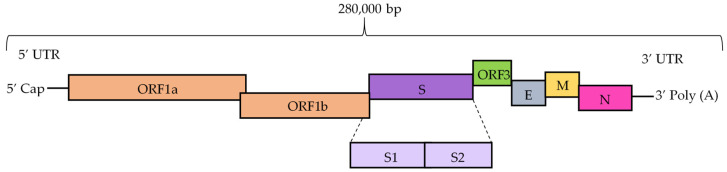
Schematic organization of PEDV genome. The genome encodes open reading frames 1a (ORF1a) and 1b (ORF1b), followed by the genes encoding spike protein (S), accessory protein 3 (ORF3), envelope protein (E), and membrane (M) and nucleocapsid (N) proteins. The genome contains untranslated regions (UTRs) at both the 5′ and 3′ termini. A poly(A) tail is present at the 3′ terminus and a 7-methyl-guanosine cap structure is present at the 5′ end.

**Figure 2 pathogens-11-01074-f002:**
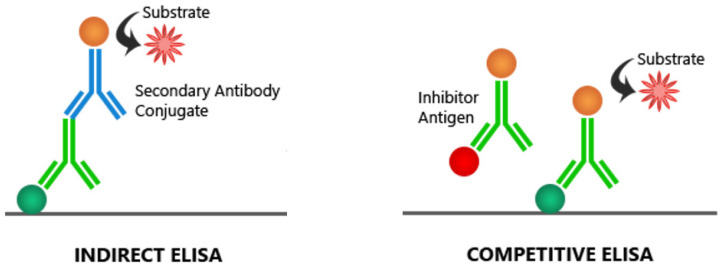
Schematic presentation of two types of ELISA used for detection of anti-PEDV antibodies.

**Figure 3 pathogens-11-01074-f003:**
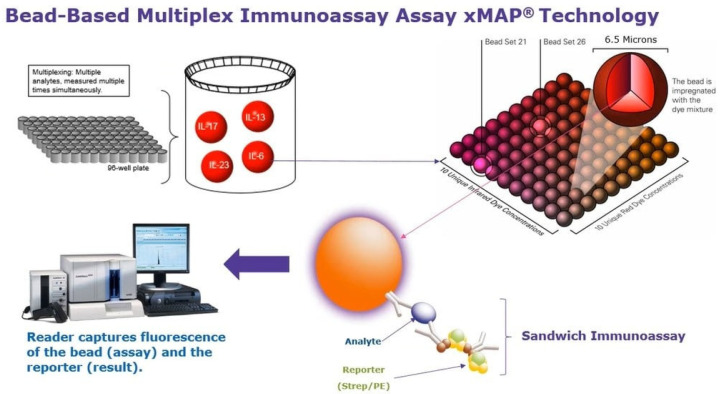
Scheme of a microsphere immunoassay (FMIA) based on Luminex^®^ xMAP^®^ technology. Luminex^®^ assays use a set of fluorescent beads where each bead falls in a unique spot on the fluorescent spectrum. The beads are coated with a capture antibody targeted to the analyte of interest. Source: https://www.sigmaaldrich.com/PL/pl/technical-documents/product-supporting/milliplex/multiplexing-in-vet-med-and-animal-health-research.

**Table 1 pathogens-11-01074-t001:** Summary of RT-PCR assays for PEDV diagnosis.

PCR Type	Primer Name	Primer Sequences (5′-3′)	Target Region (Size)	Limit of Detection	Ref.
RT-PCR	PEVD M-F PEDV M-R	ACACCTATAGGGCGCCTGTA AACCCTAAGAGGGGCATAGA	M (854 bp)	100 TCID_50_/sample	[62]
RT-nested PCR	PEDV/N-F PEDV/N-R PEDV/N-F2 PEDV/N-R2	TTGGCATTTCTACTACCTCGGA AGATGAAAAGGTACTGCGTTCC AGGAACGTGACCTCAAAGACATCCC CCAGGATAAGCCGGTCTAACATTG	N (1327 bp) N (540 bp)	Not defined	[63]
RT-PCR	PEDV Fwd PEDV Rev	ACAAGTCTCGTAACCAGTCC GTATCACCACCATCAA	N (691 bp)	Not defined	[64]
RT-PCR	PEDV P1 PEDV P2	GGACACATTCTTGGTGGTCT GTTTAGACTAAATGAAGCACTTTC	M (377 bp)	10^4^ TCID_50_/mL	[65]
RT-PCR	PEDV E1 PEDV E2	TAGACAAGCTTCAAATGTGAC GTATTAAAGATAATAAAGAGCGC	OFR3 (264 bp for field strains and 215 for attenuated strains)	6.10 × 10^4^ -7.30 copies/µL × 10^5^	[66]
RT-PCR	F1-V R1	CCAGGTGCTCAGCTAACACT TCATTATCCCATGTTATGCC	S (442 bp for variant PEDV)	5 × 10^5^ DNA copies/reaction	[67]
	F1-C R1	TCTCAGTTACATCGATTCTGG TCATTATCCCATGTTATGCC	S (270 bp for classical PEDV)	5 × 10^5^ DNA copies/reaction	
RT-PCR	PEDV F PEDV R	TTTATTCTGTCACGCCAT AGATTTACAAACACCTATGTTA	S1 (197 bp)	Not defined	[68]
RT-PCR	PEDV F PEDV R	GGTTCTATTCCCGTTGATGAGGT AACACAAGAGGCCAAAGTATCCAT	M (170 bp)	Not defined	[69]
Multiplex RT-PCR (PEDV/TGEV/GAR)	PEDV P1 PEDV P2	(TTCTGAGTCACGAACAGCCA (CATATGCAGCCTGCTCTGAA	S (651 bp)	10^2^ TCID_50_/mL	[70]
	TGEV T1 TGEV T2	GTGGTTTTGGTYRTAAATGC CACTAACCAACGTGGARCTA	S (859 bp)	10^1^ TCID_50_/mL	
	GAR rot3 GAR rot 5	AAAGATGCTAGGGACAAAATTG TTCAGATTGTGGAGCTATTCCA	Segment 6 region (309 bp)	10^1^ TCID_50_/mL	
Multiplex RT-PCR (PCV2/TGEV/PEDV/GAR)	PCV2-F PCV2-R	CGGATATTGTAGTCCTGGTCG ACTGTCAAGGCTACCACAGTC	ORF2 (481 bp)	2.17 × 10^3^/reaction	[71]
	TGEV-F TGEV-R	GTGGTTTTGGTYRTAAATG ACTAACCAACGTGGARCTA	S (859 bp)	1.74 × 10^4^/reaction	
	PEDV-F PEDV-R	TTCTGAGTCACGAACAGCCA CATATGCAGCCTGCTCTGAA	S (651 bp)	2.1 × 10^3^/reaction	
	GAR-F GAR-R	AAAGATGCTAGGGACAAAATTG TTCAGATTGTGAGCTATTCCA	Segment 6 region (309 bp)	1.26 × 10^4^/reaction	
Multiplex RT-PCR (PEDV/TGEV/PRV-A/PSaV/PKoV/PDCoV)	PEDV-F PEDV-R	TAGGACTCGTACTGAGGGTGT CTATTTTCGCCCTTGGGAATT	N (600 bp)	1 ng cDNA	[72]
	PKoV-F PKoV-R	GGCATTGACATGAATCAGGC GCGATCGTAGGTCTTCGG	Polyprotein (998 bp)	10 ng cDNA	
	TGEV-F TGEV-R	GGGCCAACGTAAAGAGCTTCC GCTCTGACCTTTCTGCAG	N (820 bp)	1 ng cDNA	
	PDCoV-F PDCoV-R	GCTGACACTTCTATTAAAC TTGACTGTGATTGAGTAG	N (497 bp)	1 ng cDNA	
	PRV-A-F PRV-A-R	GTATGGTATTGAATATACC TAGACTGATCCAGTTGGC	VP7 (350 bp)	10 ng cDNA	
	PSaV-F PSaV-R	TACAGCAAGTGGGAC ATGACACTGGTGAACGGCAT	Polyprotein (194 bp)	1 ng cDNA	
Multiplex RT-PCR (PRRSV/PEDV/CSFV/TGEV)	PEDV M-F PEDV M-R	GGTGTCAAGATGGCTATTCTATGG TGAAGCATTGACTGAACGACCA	M (435 bp)	1 × 10^3^ copies/µL	[73]
	CSFV 5′UTR-F CSFV 5′UTR-R	GCTCCCTGGGTGGTCTAAGTC GGGTTAAGGTGTGTCTTGGGC	5′UTR (116 bp)	1 × 10^3^ copies/µL	
	PRRSV M-F PRRSV M-R	ACCTCCAGATGCCGTTTGTG GCTTTTCTGCCACCCAACAC	M (197 bp)	1 × 10^3^ copies/µL	
	TGEV N-F TGEV N-R	GACAAACTCGCTATCGCATGG AGTGGTATTTGTGTGTGAACGTGA	N (720 bp)	1 × 10^3^ copies/µL	
Multiplex nested RT-PCR (PEDV/TGEV/PRV-A)	PEDV M1-FPEDV M2-RPEDV M3-FPEDV M4-R	GAATTTTACATGGAATATCATACTGACGATACTACTTGT CGCCAGTAGCAACCTTATAGCCCTCTA TGCTTCAGTATGGCCATTACAAGTACTCTG CCTGTCGGCCCATCACAGAAGTAGT	M (450 bp) M (291 bp)	27.2 µg/µL RNA	[74]
	TGEV S1 TGEV S2 TGEV S3 TGEV S4	AGGGTAAGTTGCTATTAGAAATAATGGTAAGTT CTAATTTACCACTAACCAACGTGGAGCTATTA AAAAATTATTTGTGGTTTTGGTTGTAATGCC GTGTAGTAAAAACATTAGCCACATAACTAGCACA	S (950 bp) S (792 bp)	10^2^ TCID_50_/mL	
	PRV-A P1 PRV-A P2 PRV-A P3 PRV-A P4	GGCTTTTAAAGCGCTACAGTGATGTCTCT GGTCGTGATTGTGTTGATGAATCCATAGA CTCAGCATTGACGTAACGAGTCTTCC TGAGTGGATCGTTTGAAGCAGAATCAGA	NSP5 (317 bp) NSP5 (208 bp)	10^1^ TCID_50_/mL	
One-step triplex RT-PCR (PEDV/PSV/PSaV)	PEDV-F PEDV-R	CTGCCAATGTATTTGCCAC GGAAGTTCCTTGAACCTCG	S1 (659 pb)	10^4^ copies/µL	[75]
	PSV-F PSV-R	TGCTTGAGGAGTCGGAGAG GCCCTGCACAACTGCTTTC	Conserved region (428 bp)	10^4^ copies/µL	
	SaV-F SaV-R	TACGGGGGAATAGGTTT CAGCCACATCTGGGTAGT	VP1 (246 bp)	10^4^ copies/µL	
Duplex RT-PCR (TGEV/PEDV)	PEDV T1 PEDV T2	GTGGTTTTGGTYRTAAATGC CACTAACCAACGTGGARCTA	S (651 bp)	10^1^ TCID_50_/mL	[76]
	TGEV P1 TGEV P2	TTCTGAGTCACGAACAGCCA CATATGCAGCCTGCTCTGAA	S (859 bp)	10^2^ TCID_50_/mL	
Multiplex RT-PCR (PEDV/TGEV)	PEDV PA PEDV PB	GGGCGCCTGTATAGAGTTTA AGACCACCAAGAATGTGTCC	M (412 bp)	10 TCID_50_/mL	[77]
	TGEV TA TGEV TB	GATGGCGACCAGATAGAAGT GCAATAGGGTTGCTTGTACC	N (612 bp)	10 TCID_50_/mL	

F—forward; R—reverse; PEDV—porcine epidemic diarrhea virus; TGEV—transmissible gastroenteritis virus; PSV—porcine sapelovirus; PSaV—porcine sapovirus; PRV-A—porcine group A rotavirus; PRRSV—porcine respiratory syndrome virus; CSFV—classical swine fever virus; PKoV—porcine kobuvirus; PDCoV—porcine deltacoronavirus; GAR—porcine group A rotavirus; PCV2—porcine circovirus 2; Ref.—reference’ bp- base pair.

**Table 3 pathogens-11-01074-t003:** The advantages and disadvantages of serological assays and the nucleic acid test for PEDV diagnostic purposes.

	Method	Advantages	Disadvantages
Serological tests	ELISA	High sensitivity and specificity, multiple formats available, not very expensive, medium turnaround time, IgG or IgA antibody detection and amenable to high-volume testing	Needs infrastructure and qualified personnel, and nonspecific result can occur
	Antigen ELISA	High sensitivity and specificity, amenable to high-volume testing, not very expensive and medium turnaround time	Needs infrastructure and qualified personnel, nonspecific reaction can occur and fecal samples should be collected immediately after appearance of clinical symptoms
	FMIA	Sensitive, specific, rapid, simultaneous detection of antibodies against multiple pathogens and amenable to high-volume testing	Expensive, and requires specialized equipment, reagents and qualified personnel
	VN	High specificity, detected neutralizing antibodies and not very expensive	Time-consuming, requires well-trained technicians, subjective results, cytotoxic effects can occur and requires Vero cell culture
	FFN	High specificity, cytopathic effect is not interpreted, shorter turnaround time than VN, can be quantitative or qualitative and detected neutralizing antibodies	Requires specialized equipment, reagents and qualified personnel, and requires Vero cell culture
	IFA	High specificity and medium turnaround time	Requires specialized equipment, reagents and qualified personnel; lower diagnostic sensitivity than FFN; subjective results; and requires Vero cell culture
	IC	No need for laboratory equipment and qualified personnel, portable, rapid turnaround, speedy results, simple test procedure and relatively inexpensive to produce	Low specificity and sensitivity, most IC assays are not quantitative and are not suitable for large-scale screening, one step assay and nonspecific results can occur
Nucleic acid tests	RT-PCR/nRT-PCR	Higher sensitivity and specificity, and can be multiplexed	Expensive; requires specialized equipment, reagents and qualified personnel; time-consuming; unable to quantify the target DNA; only qualitative test; and not suitable for large-scale screening
	qRT-PCR	Higher sensitivity and specificity, more sensitive than RT-PCR, quantitative application, medium turnaround time, can be multiplexed and amenable to high-volume testing	Expensive; requires specialized equipment, reagents and qualified personnel; and needs quality control
	LAMP	Simple, fast, cheap, no special equipment, high sensitivity, can be quantitative or qualitative, results can be read by eye and the temperature stability of the reagents enables its use in field condition	Requires rather qualified personnel, primer design can be challenging, nonspecific binding of primers can cause false-positive results, and for qualitative results, specialized equipment is required
	Sequencing	High accuracy, genotyping, provides the opportunity to detect recombination events, NGS has the ability to diagnose entire genome and discover novel, unknown PEDV strains, and using NGS, there is no need for target-specific primers	Expensive; requires specialized equipment; reagents and bioinformatics experts; and difficult for widespread testing
	CRISPR	Higher specificity and sensitivity, accuracy, rapid, can be used alone or with established amplification methods and can be multiplexed	Generally, requires pre-amplification step and qualitative detection

ELISA—enzyme-linked immunosorbent assay; FMIA—fluorescent microsphere immunoassay; VN—virus neutralization assay; FFN—fluorescent focus neutralization; IFA—indirect immunofluorescence assay, IC—immunochromatography assay; RT-PCR—reverse-transcription polymerase chain reaction; nRT-PCR—nested reverse-transcription polymerase chain reaction; qRT-PCR—real-time reverse-transcription polymerase chain reaction; LAMP—loop-mediated isothermal amplification assays; CRISPR—clustered regularly interspaced short palindromic repeats.

## Data Availability

Not applicable.

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
