# Peer review of "Current State of Molecular and Serological Methods for Detection of Porcine Epidemic Diarrhea Virus"

_pathogens, 2022, doi:10.3390/pathogens11101074_

Round 1

Reviewer 1 Report

Dear editor,

The manuscript entitled “Current States of molecular and serological methods for detection of porcine epidemic diarrhea virus” is a review that the author intended to be used by other scientists for having a good over view on the current methods and techniques used in diagnosis.

With the fast development of science in the past 20 years, overviews became so important in all fields to sum-up new discoveries and information for scientists.

Over all, the review is good; I think it can be move developed by adding more examples about those methods. Because the review states in general those methods, but it even misses figures and graphs describing those methods.

Although, I have some points and comments that I hope the author would address.

Abstract:

In the text the author keeps on using RT-PCR and Real-time PCR, I would recommend the following, since this can be so complicated for some readers:

Use RT-PCR for reverse transcription PCR

Use qPCR for real-time PCR

And qRT-PCR for real-time reverse transcription PCR

This is the case with all RNA viruses, because RT-PCR sometimes can be used for real-time PCR, that’s why we recommend qPCR and qRT-PCR.

line 20: PEDV DNA, please change to cDNA because PEDV is an RNA virus, and what we synthsize in the laboratory is a cDNA not a DNA.

Clustered regulatory interspaced short palindrome repeats please provide the abbreviation (CRISPR)

Introduction:

Line 41: 280,000 base pairs, please correct it to 28,000 bp.

Line 59: I think here you can provide a figure explaining the construction of the genome, since the genome is so complicated and it would be easier for any leader to see the figure with all the names and order of ORFs and how some ORF are further divided.

Line 61-70 please provide more references, because you only have one reference for this part which is number 8 (Yu J. et., 2018) which doesn’t mention these information.

Line 321: Please rename to Methods for the detection of PEDV genome and/or antigens.

All PCR methods detect the genome and not the virus itself.

Line 354: cDNA instead of DNA

Line 502: CRISPR not CRISP

Author Response

We would like thank the reviewer for his comments on our manuscript. We have acted upon the suggestions provided by the reviewer  and alterations were included in the updated version of the manuscript.

Reviewer 1

The manuscript entitled “Current States of molecular and serological methods for detection of porcine epidemic diarrhea virus” is a review that the author intended to be used by other scientists for having a good over view on the current methods and techniques used in diagnosis.

With the fast development of science in the past 20 years, overviews became so important in all fields to sum-up new discoveries and information for scientists.

Over all, the review is good; I think it can be move developed by adding more examples about those methods. Because the review states in general those methods, but it even misses figures and graphs describing those methods.

Re: The tables showing more examples and figures have been included.

Although, I have some points and comments that I hope the author would address.

  1. Abstract:

In the text the author keeps on using RT-PCR and Real-time PCR, I would recommend the following, since this can be so complicated for some readers:

Use RT-PCR for reverse transcription PCR. Use qPCR for real-time PCR. And qRT-PCR for real-time reverse transcription PCR

This is the case with all RNA viruses, because RT-PCR sometimes can be used for real-time PCR, that’s why we recommend qPCR and qRT-PCR.

Re: It has been corrected

line 20: PEDV DNA, please change to cDNA because PEDV is an RNA virus, and what we synthsize in the laboratory is a cDNA not a DNA.

Re: It has been corrected

Clustered regulatory interspaced short palindrome repeats please provide the abbreviation (CRISPR)

Re: It has been corrected

  1. Introduction:

Line 41: 280,000 base pairs, please correct it to 28,000 bp.

Re: It has been corrected

Line 59: I think here you can provide a figure explaining the construction of the genome, since the genome is so complicated and it would be easier for any leader to see the figure with all the names and order of ORFs and how some ORF are further divided.

Re: The figure has been included.

Line 61-70 please provide more references, because you only have one reference for this part which is number 8 (Yu J. et., 2018) which doesn’t mention these information.

Re: It has been corrected

Line 321: Please rename to Methods for the detection of PEDV genome and/or antigens.

Re: It has been corrected

All PCR methods detect the genome and not the virus itself.

Re: Of course I agree with the reviewer. It has been corrected

Line 354: cDNA instead of DNA

Re: It has been corrected

Line 502: CRISPR not CRISP

Re: It has been corrected

Reviewer 2 Report

Dear colleagues,

Thank you very much for such interesting reading

The structure and language of the manuscript is clear and easily understandable. The only thing I would mention is about the methodology used. To dispel any doubts that the literature selection methodology is correct, and the publications overviewed in this manuscript cover the full range of them, to ensure that nothing is omitted, it would be useful to add a description of the selection methodology.

From a practical point of view (at least in my view), it would be also useful at the end to summarize (for instance as a table) which diagnostic method (or combination) is preferable to use for certain purposes, in certain conditions.

Author Response

We would like thank the reviewer for his comments on our manuscript. We have acted upon the suggestions provided by the reviewer  and alterations were included in the updated version of the manuscript.

Reviewer 2

Thank you very much for such interesting reading

The structure and language of the manuscript is clear and easily understandable. The only thing I would mention is about the methodology used. To dispel any doubts that the literature selection methodology is correct, and the publications overviewed in this manuscript cover the full range of them, to ensure that nothing is omitted, it would be useful to add a description of the selection methodology.

Re: The methodology has been included however I would like to point that this review is non-systematic review.

From a practical point of view (at least in my view), it would be also useful at the end to summarize (for instance as a table) which diagnostic method (or combination) is preferable to use for certain purposes, in certain conditions.

Re: the table has been included.